# Spatiotemporal-resolved protein networks profiling with photoactivation dependent proximity labeling

Yansheng Zhai[1,5], Xiaoyan Huang[1,5], Keren Zhang[1], Yuchen Huang[1], Yanlong Jiang[2], Jingwei Cui[1], Zhe Zhang[1,3], Cookson K. C. Chiu[4], Weiye Zhong[1] & Gang Li [1] ✉

Enzymatic-based proximity labeling approaches based on activated esters or phenoxy radicals have been widely used for mapping subcellular proteome and protein interactors in living cells. However, activated esters are poorly reactive which leads to a wide labeling radius and phenoxy radicals generated by peroxide treatment may disturb redox-sensitive pathways. Herein, we report a photoactivation-dependent proximity labeling (PDPL) method designed by genetically attaching photosensitizer protein miniSOG to a protein of interest. Triggered by blue light and tunned by irradiation time, singlet oxygen is generated, thereafter enabling spatiotemporally-resolved aniline probe labeling of histidine residues. We demonstrate its high-fidelity through mapping of organelle-specific proteomes. Side-by-side comparison of PDPL with TurboID reveals more specific and deeper proteomic coverage by PDPL. We further apply PDPL to the disease-related transcriptional coactivator BRD4 and E3 ligase Parkin, and discover previously unknown interactors. Through over-expression screening, two unreported substrates Ssu72 and SNW1 are identified for Parkin, whose degradation processes are mediated by the ubiquitination-proteosome pathway.

The precise characterization of protein networks underpins many fundamental cellular processes[1]. Thus, high-fidelity spatiotemporal mapping of protein interactions would provide a molecular basis for deciphering biological pathways, disease pathologies, as well as the potential to perturb these interactions for therapeutic purposes[2]. To this end, methods that can detect transient interactions in living cells or tissues are highly needed. Affinity purification-mass spectrometry (AP-MS) was historically used to pull down binding partners of a protein of interest (POI). With advances in quantitative proteomics approaches, the largest protein network database Bioplex3.0 based on AP-MS has been established[3]. While AP-MS is extremely powerful, the cell lysis and dilution steps in the workflow bias against weak and

transient binding interactions and introduce the post-lysis artefacts, such as spuriously interacting pairs lacking compartmentalization prior to lysis[4].

In attempts to tackle these challenges, unnatural amino acids (UAA) with crosslinking groups and enzymatic proximity labeling (PL) platforms (e.g., APEX and BioID) have been developed[5]. While UAA methods have been successfully applied in many scenarios and provide information on direct protein binders[6], it still requires optimization of the UAA insertion sites. More importantly, it is a stoichiometric labeling method lacking catalytic turnover of the labeling event. Conversely, enzymatic PL methods such as the BioID method fuse an engineered biotin ligase to the POI[7], with subsequent activation of biotin to

[1]Institute of Systems and Physical Biology, Shenzhen Bay Laboratory, Shenzhen 518132, China. [2]Department of Chemistry, UF Scripps Biomedical Research, 130 Scripps Way, Jupiter, FL 33458, USA. [3]School of Life Sciences, University of Science and Technology of China, Hefei, Anhui 230026, China. [4]Multi-omics Mass Spectrometry Core, Office of Core Facility, Shenzhen Bay Laboratory, Shenzhen 518132, China. [5]These authors contributed equally: Yansheng Zhai, Xiaoyan Huang. ✉e-mail: ligang@szbl.ac.cn

generate the reactive ester biotinoyl-AMP intermediate. As such, the enzyme catalyzes and releases a "cloud" of activated biotin, which tags proximal lysine residues. However, BioID requires over 12 h to gain enough labeling signal, which prevents its application in a temporally-resolved manner. Using yeast display-based directed evolution, TurboID was engineered from BioID with much greater efficiency, achieving effective biotin labeling within 10 min[8], making it possible to study more dynamic processes. Since TurboID is highly active and endogenous levels of biotin are sufficient for low-level labeling, background labeling becomes a potential concern when strongly enhanced and time-regulated labeling is required through the addition of exogenous biotin. Moreover, the activated ester species are poorly reactive ($t_{1/2}$ ~5 min), which can lead to a wide labeling radius, particularly after the saturation of neighboring proteins with biotin[5]. In a different approach, a genetic fusion of engineered ascorbic acid peroxidase (i.e., APEX) generates biotin-phenol radicals upon activation with $H_2O_2$ and achieves protein labeling within a minute[9,10]. APEX has been widely used in identifying subcellular proteomes, membrane protein complexes, and cytosolic signaling protein complexes[11,12]. However, the requirement for high concentrations of peroxide may affect redox-sensitive proteins or pathways, perturbing cellular processes.

Therefore, new methods that can generate more reactive species to restrain the labeling radius with high spatial and temporal fidelity and without significant perturbation to cellular pathways would be an important complement to current methods. Among the reactive species, singlet oxygen aroused our attention due to its short lifetime and limited diffusion radius ($t_{1/2} < 0.6$ μs in cells)[13]. Singlet oxygen has been reported to promiscuously oxidize methionine, tyrosine, histidine, and tryptophan to umpolung their polarity[14,15], which are ligated with amine or thiol-based probes[16,17]. Even though singlet oxygen has been applied in RNA labeling in subcellular compartments[18], repurposing the strategy in proximity labeling of endogenous POIs remains untapped. Herein, we present a platform named photoactivation-dependent proximity labeling (PDPL), where we use blue light to illuminate a photosensitizer miniSOG-fused POI and trigger singlet oxygen generation to oxidize proximal residues, followed by modification of oxidized intermediates with an amine-containing chemical probe in live cells. We screen a panel of chemical probes to maximize the labeling specificity and determine the modification sites using an open search proteomics workflow. A side-by-side comparison of PDPL with TurboID reveals more specific and deeper proteomic coverage by PDPL. We apply this approach to organelle-specific labeling of subcellular proteomes and proteome-wide identification of binding partners for the cancer-related epigenetic regulator protein BRD4 and Parkinson's disease-related E3 ligase Parkin, which validates a network of known and unreported protein interactors. The ability of PDPL to identify E3 substrates within the large size of protein complexes represents a scenario where the identification of indirect binders is required. Two unreported Parkin substrates mediated by the ubiquitination-proteosome pathway are validated in situ.

## Results

### Development of the PDPL platform
Photodynamic therapy (PDT)[19] and chromophore-assisted laser inactivation (CALI)[20], where singlet oxygen is produced by photoirradiation of the photosensitizer, are able to inactivate target proteins or trigger cell death. As singlet oxygen is a highly reactive species with a theoretical diffusion distance of ~70 nm[17,18,21], spatially restricted oxidation can be controlled around the photosensitizer. Based on this concept, we decided to take advantage of singlet oxygen to achieve proximity labeling of protein complexes in live cells. We designed the PDPL chemical proteomic method to embody four features: (1) catalytic generation of reactive singlet oxygen in a similar fashion as enzymatic PL approaches; (2) provide temporally-resolved labeling by initiation through light illumination; (3) allow spatially-resolved

labeling by altering the irradiation time to tune the labeling radius; (4) avoid the use of endogenous cofactors (e.g., biotin) to reduce background or the use of highly perturbative exogenous reagents (e.g., peroxides) to minimize impact to the cellular environment.

Photosensitizers can be divided into two categories, including small molecule-based fluorophores (e.g., Rose Bengal, Methylene Blue)[22] and genetically-encoded small proteins (e.g., miniSOG, KillerRed)[23]. To enable a modular design, we developed the first generation PDPL platform by appending a photosensitizer (PS) protein[24,25] to the POI (Fig. 1a). After blue light illumination, singlet oxygen oxidizes proximal nucleophilic amino acid residues resulting in umpolung polarity, which is electrophilic and can further react with an amine probe nucleophile[16,17]. The probes are designed with an alkyne handle, enabling click chemistry and pull-down for LC-MS/MS characterization.

We started by testing the well-developed photosensitizers miniSOG[26] and KillerRed[23] stably expressed in HEK293T for their capacity to mediate proteomic labeling, with propargyl amine as the chemical probe (Supplementary Fig. 1a). In-gel fluorescence analysis revealed that proteome-wide labeling was achieved with miniSOG and blue light illumination while no obvious labeling products were observed for KillerRed. To increase the signal-to-background ratio, we next tested a panel of chemical probes, containing either aniline (1 and 3), propyl amine (2), or benzylamine (4). We noticed HEK293T cells alone have a higher background signal relative to the omission of blue light, presumably due to the endogenous photosensitizer riboflavin, flavin mononucleotide (FMN)[27]. Aniline-based chemical probes 1 and 3 gave better specificity, with HEK293T stably expressing miniSOG in mitochondria displaying an >8-fold increase in signal for probe 3, while probe 2 used in the RNA-labeling method CAP-seq only displaying ~2.5-fold signal increase, likely due to different reactivity preferences between RNA and protein (Fig. 1b, c). Furthermore, the isomers of probe 3 and hydrazine probe (probe 5, 6, 7) were also tested, confirming probe 3 as the optimized one (Supplementary Fig. 1b, c). Likewise, in-gel fluorescence analysis determined other optimized experimental parameters: illumination wavelength (460 nm), chemical probe concentration (1 mM), and illumination time (20 min) (Supplementary Fig. 2a–c). Omission of any component or step in the PDPL protocol resulted in significant reversion of signal-to-background (Fig. 1d). Notably, protein labeling was greatly reduced in the presence of sodium azide or trolox, which are known to quench singlet oxygen[28]. The presence of $D_2O$, known to stabilize singlet oxygen, enhanced the labeling signal. To investigate the contribution of other reactive oxygen species to labeling, mannitol and vitamin C, established hydroxyl radical and superoxide radical scavengers respectively[18,29], were added but not found to reduce labeling. The addition of $H_2O_2$ rather than illumination failed to generate labeling (Supplementary Fig. 3a). Fluorescent imaging of singlet oxygen by Si-DMA probe confirmed the presence of singlet oxygen in the HEK293T-miniSOG line, but not the parental HEK293T line. In addition, mitoSOX Red was unable to detect superoxide generation after illumination (Fig. 1e and Supplementary Fig. 3b)[30]. These data strongly indicate singlet oxygen as the major ROS that gives rise to subsequent proteome labeling. The cytotoxicity of PDPL, including the blue light illumination and chemical probe treatment, was evaluated, and no significant cytotoxicity was observed (Supplementary Fig. 4a).

### Characterization of PDPL labeling sites and validation of labeling in vitro
To explore the labeling mechanism and achieve proteomic identification of protein complexes by LC-MS/MS, we first needed to determine which amino acids were modified and the delta mass of probe labeling. Methionine, histidine, tryptophan, and tyrosine have been reported to be modified by singlet oxygen[14,15]. We integrated the TOP-ABPP

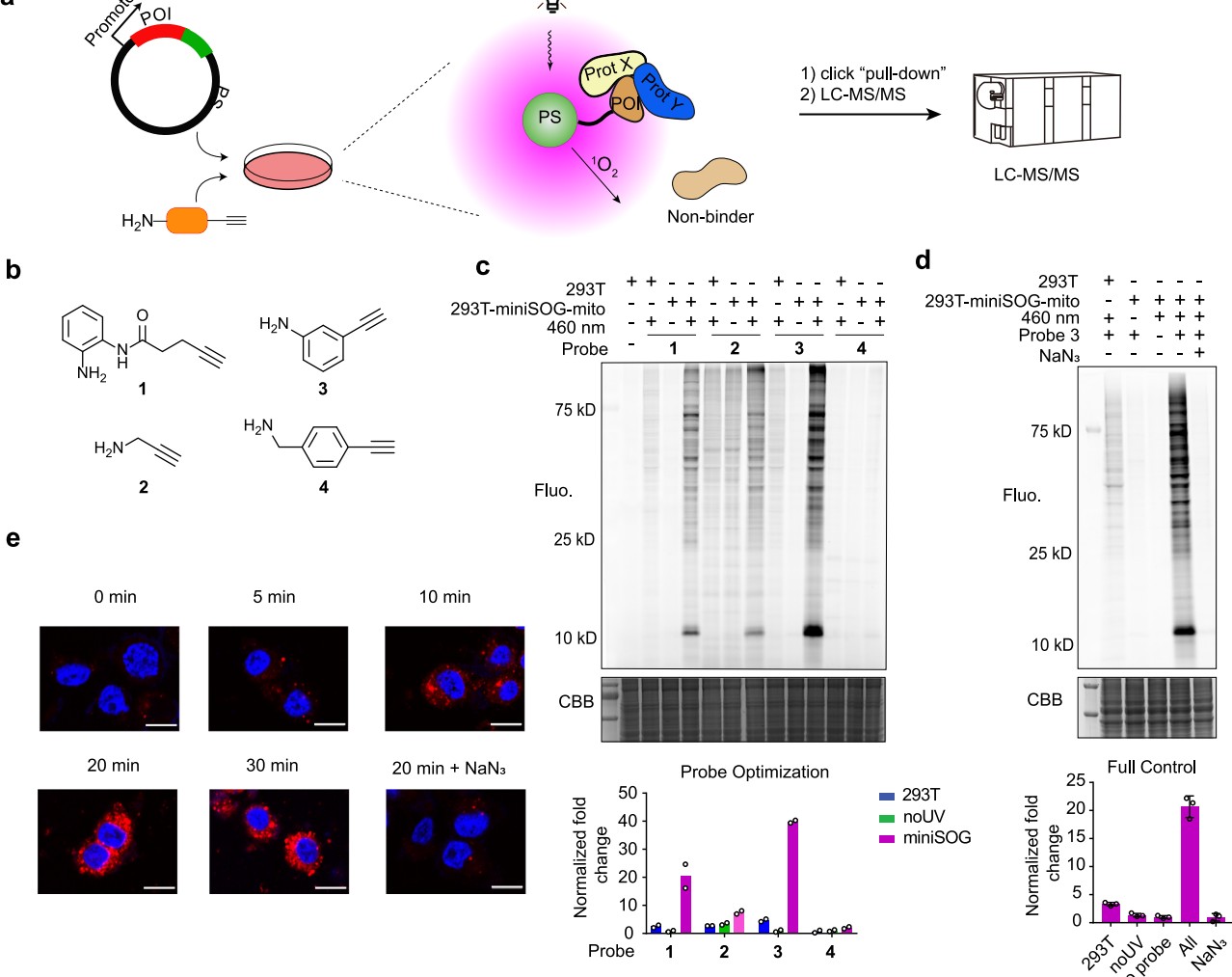

**Fig. 1 | Development and characterization of photoactivation-dependent proximity labeling method. a** Schematic of miniSOG-mediated protein complex labeling. On blue light illumination, miniSOG-POI-expressing cells generate singlet oxygen, which modifies interacting proteins rather than non-binder proteins. The photo-oxidation intermediate is intercepted by amine chemical probe relay labeling to form a covalent adduct. The alkyne group on the chemical probe enables click chemistry for pull-down enrichment, followed by quantitative analysis by LC-MS/MS. **b** The chemical structure of amine probes **1-4**. **c** Representative fluorescent gel analysis of mitochondria-localized miniSOG-mediated proteomic labeling with probes **1-4** as well as the relative quantification based on gel densitometry. Negative control experiments omitting blue light or with HEK293T cells without miniSOG expression were used to evaluate the signal-to-background labeling of the chemical probes. $n = 2$ biologically independent samples. Each dot represents a biological replicate. **d** Representative PDPL detection and quantification using optimized probe **3** in the presence or absence of the indicated PDPL components similar to **c**. $n = 3$ biologically independent samples. Each dot represents a biological replicate. Center line and whiskers denote the mean and ±SD. CBB: coomassie brilliant blue. **e** Confocal imaging of singlet oxygen by far-red dye Si-DMA. Scale bar: 10 μm. Gel imaging and confocal experiments were independently repeated at least twice with similar results.

workflow[31] with unbiased open search enabled by the MSFragger-based FragPipe computational platform[32]. After singlet oxygen modification and chemical probe labeling, a cleavable linker-containing biotin retrieval tag was used for click chemistry, followed by neutravidin pull-down and trypsin digestion. Modified peptides, still bound on resin, were photo-cleaved for LC-MS/MS analysis (Fig. 2a and Supplementary Data 1). Masses of modification that occur proteome-wide with over 50 peptide spectrum matches (PSMs) are listed (Fig. 2b). Surprisingly, we only observed modification on histidine, presumably due to the high reactivity of oxidized histidine toward aniline probe over other amino acids. According to the published mechanism of histidine oxidation by singlet oxygen[21,33], the putative structure for delta mass +229 Da corresponds to the adduct for probe **3** with 2-oxo-histidine after twice oxidation, whereas +247 Da is the hydrolysis product of +229 Da (Supplementary Fig. 5). Evaluation of MS2 spectra show high confidence identification of a large fraction of y

ions and b ions, including fragment ions (y and b) that allowed identification of the modification (Fig. 2c). Analysis of the local sequence context of PDPL-modified histidines revealed a modest motif preference for small, hydrophobic residues at the ±1 position (Supplementary Fig. 4b). On average, 1.4 histidines per protein were identified and these labeled sites are well exposed determined by solvent-accessible surface area (SASA) and relative solvent accessibility (RSA) analysis (Supplementary Fig. 4c, d).

To biochemically validate the labeling sites, the histidines of PRDX3 and PRDX1 identified in mass spectrometry were mutated to alanine and compared with wildtype in a transfection assay. PDPL results showed the mutations significantly reduce labeling (Fig. 2d). Meanwhile, a peptide sequence identified in the open search was synthesized and reacted in vitro with purified miniSOG in the presence of probe **3** and blue light, with products with mass shift +247 and +229 Da appearing in LC-MS detection (Fig. 2e). To test if proximal

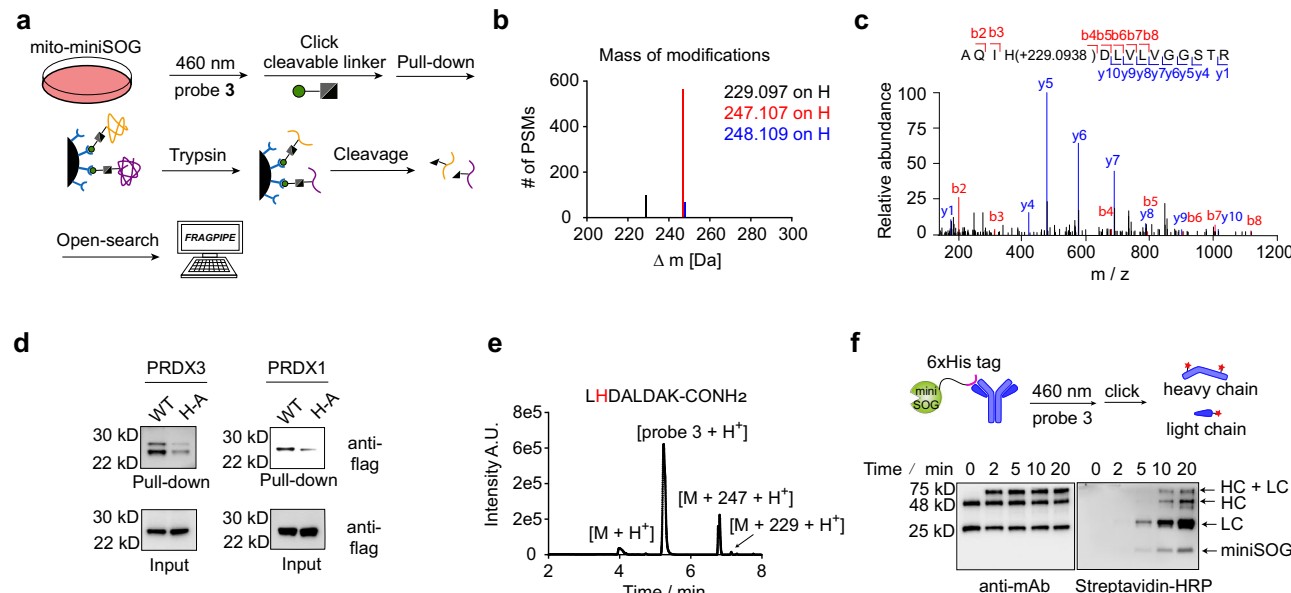

**Fig. 2 | Characterization of PDPL labeling sites and validation of labeling in vitro. a** An unbiased workflow to study residue selectivity using the MSFragger-based FragPipe computational platform. A cleavable linker was used for Click chemistry, which enables the photo-cleavage of the modified peptide from the streptavidin resin. An open search was deployed to identify the masses of modification as well as the corresponding residues. **b** The masses of modification that occur proteome-wide are assigned. PSMs peptide spectrum matches. **c** MS2 spectrum annotation of a probe **3**-modified histidine site. Covalent reaction with probe 3 adds +229.0938 Da to the modified amino acid as a representative example. **d** Mutation assay to validate PDPL labeling. PRDX3(H155A, H225A) and

PRDX1(H10A, H81A, H169A) were transfected together with their wild-type plasmids for anti-Flag detection. **e** A synthetic peptide was reacted with purified miniSOG in the presence of probe **3** and the corresponding product with Δm +247 and +229 was annotated in the LC-MS spectrum. **f** In vitro protein-protein interaction simulated by miniSOG-6xHis-tag and anti-6xHis antibody. Anti-biotin (streptavidin-HRP) and anti-mouse Western blot analysis of probe **3**-labeled miniSOG-6xHis/anti-6xHis antibody complex against light irradiation time. Labeling of individual proteins are indicated at appropriate molecular weights: LC light chain of antibody; HC heavy chain of the antibody. These experiments were independently repeated at least twice with similar results.

protein interactors could be labeled in response to the photoactivation of miniSOG in vitro, we constructed an artificial proximity assay via the interaction between miniSOG-6xHis protein and anti-His monoclonal antibody in vitro (Fig. 2f). In this assay, we expected proximal labeling of both heavy and light chains of the antibody by miniSOG. Indeed, anti-mouse (recognizing both heavy- and light-chains of anti-6xHis tag antibody) and streptavidin Western blot revealed robust biotinylation of heavy and light chains. Notably, we noticed self-biotinylation of miniSOG due to the 6xHis tag as well as crosslinking between light and heavy chain, presumably due to the proximal reaction between lysine and 2-oxo-histidine, which has been previously reported[34]. Taken together, we concluded that PDPL modifies histidine in a proximity-dependent manner.

### Identification of organelle-specific proteomes by PDPL

Our next aim was to characterize subcellular proteomes to test in situ labeling specificity. We, therefore, stably expressed miniSOG in the nucleus, mitochondrial matrix or ER outer membrane of HEK293T cells (Fig. 3a). In-gel fluorescence analysis revealed abundant labeling bands as well as different labeling patterns across the three subcellular locations (Fig. 3b). Fluorescent imaging analysis revealed high specificity of PDPL (Fig. 3c). The PDPL workflow followed by click reaction with a rhodamine dye was used to delineate the subcellular proteomes by fluorescence microscopy, with PDPL signal co-localizing with DAPI, mitochondrial tracker, or ER tracker, validating high fidelity of PDPL. For three organelle locations, a side-by-side comparison of PDPL with TurboID using anti-biotin Western blot revealed more specific labeling by PDPL compared with their respective controls. More labeling bands appeared in PDPL conditions, indicating more labeled proteins by PDPL (Supplementary Fig. 6a–d).

Encouraged by the gel and imaging results, label-free quantification was employed to quantify the identified proteomes in each

organelle (Supplementary Data 2). Non-transfected HEK293T was used as the negative control to deduct the background labeling. Volcano plot analysis displayed significantly enriched proteins ($p < 0.05$ and >2-fold ion intensity) as well as singleton proteins that are only present in miniSOG-expressing lines (Fig. 3d red and green dots). Combining these data, we identified 1364, 461, and 911 statistically significant proteins for the nucleus, mitochondria, and ER outer membrane, respectively. To analyze the accuracy of organelle-localized PDPL, we used MitoCarta 3.0, Gene ontology (GO) analysis, and the A. Ting et al. dataset[8] for mitochondria, nucleus, and ER to validate the organelle-specificity of detected proteins, which corresponded to 73.4, 78.5, and 73.0% accuracy (Fig. 3e). The specificity of PDPL validates that PDPL is an ideal tool for identifying organelle-specific proteomes. Notably, submitochondrial analysis of the identified mitochondrial proteins revealed that the captured proteome was mainly distributed in the matrix and inner membrane (226 and 106, respectively), accounting for 91.7% of the total identified mitochondrial proteins (362), which further confirmed the high-fidelity of PDPL (Supplementary Fig. 7a). Likewise, the subnuclear analysis revealed the captured proteome was predominantly distributed in the nucleus, nucleoplasm, and nucleolus (Supplementary Fig. 7b). Nucleus proteome profiling by nuclear localization signals (3xNLS) peptide revealed similar accuracy to H2B construct (Supplementary Fig. 7c−h). To define the labeling specificity of PDPL, nuclear Lamin A was selected as a more discretely localized POI bait[7]. PDPL identified 36 significantly enriched proteins, with 12 proteins (30.0%, including Lamin A) being well-characterized Lamin A interacting proteins annotated by the String database, representing a higher percentage than the BioID method (28 out of 122 proteins, 22.9%)[7]. Our method identified less proteins, likely due to the restricted labeling area, enabled by more reactive singlet oxygen. GO analysis reveals that the identified proteins are mainly located in the nucleoplasm (26), nuclear envelope (10), nuclear membrane (9), and nuclear

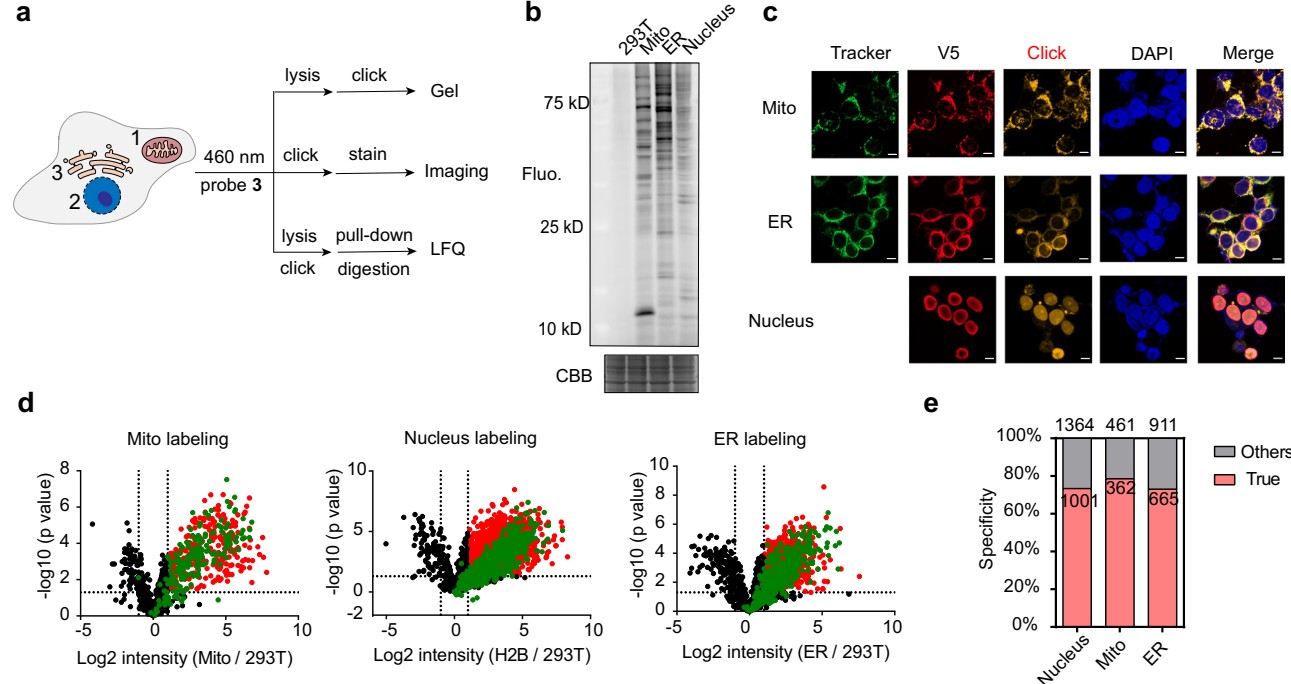

**Fig. 3 | PDPL allows for subcellular proteomic profiling in mitochondria, ER, and nucleus. a** Schematic of miniSOG-mediated organelle-specific proteomic labeling. miniSOG is genetically targeted to the mitochondrial matrix via fusion to the N-terminal 23 amino acids of human COX4 (mito-miniSOG), to the nucleus via fusion to H2B (nucleus-miniSOG), and to the cytoplasmic side of the ER membrane via fusion to Sec61β (ER-miniSOG). Readout includes gel imaging, confocal imaging and mass spectrometry. **b** Representative gel imaging of three organelle-specific PDPL profiles. CBB coomassie brilliant blue. **c** Representative confocal imaging of HEK293T cells stably expressing different subcellular-localized miniSOG, which were detected with V5 tag antibody (red). Subcellular markers are used for mitochondria and ER (green). PDPL workflow involves click chemistry with Cy3-azide to detect the miniSOG-labeled subcellular proteomes (yellow). Scale bar: 10 μm.

**d** Volcano plots of PDPL-labeled proteomes in different organelles quantified by label-free quantification (*n* = 3 independent biological experiments). A two-sided student's *t*-test was used in the volcano plot. Wild-type HEK293T was used as the negative control. Significantly changed proteins are highlighted in red (*p* < 0.05 and >2-fold ion intensity difference). Relevant proteins that are significant in HEK293T-miniSOG but not in HEK293T are marked in green. **e** Specificity analysis for proteomic data sets derived from experiments in **d**. Total numbers of statistically significant proteins in each organelle (red and green dots) were labeled on top. Bars show organelle-localized proteins based on MitoCarta 3.0, GO analysis, and A. Ting et. al. dataset for mitochondrial, nucleus, and ER, respectively. These experiments were independently repeated at least twice with similar results. Source data are provided as a Source Data file.

pore (5). Combined, these nucleus-localized proteins account for 80% of the enriched proteins, further demonstrating the specificity of PDPL (Supplementary Fig. 8a–d).

**Spatiotemporally-resolved profiling of BRD4 binding proteins**
Having established the ability of PDPL for proximity labeling in organelles, we next tested whether PDPL could be used to profile the binding partners of a POI. In particular, we sought to determine the PDPL profiling of cytosolic proteins, which were regarded as more difficult targets compared with membrane-localized counterparts due to their highly dynamic nature[12]. The bromodomain and extraterminal (BET) protein BRD4 aroused our attention for its critical role in a variety of diseases[35,36]. The complexes formed by BRD4 are transcriptional coactivators and important therapeutic targets. By regulating the expression of transcription factor c-myc and Wnt5a, BRD4 is ascribed as a key determinant in acute myeloid leukemia (AML), multiple myeloma, Burkitt's lymphoma, colon cancer, and inflammatory diseases[37,38]. In addition, several viruses target BRD4 to regulate viral and cellular transcription, such as papillomavirus, HIV and SARS-CoV-2[36,39].

To determine an interaction map of BRD4 using PDPL, we fused miniSOG to the short isoform of BRD4 at either N- or C-terminal. Proteomic results revealed a high overlap between the two constructs (Supplementary Fig. 9a). The nuclear proteome determined by miniSOG-H2B covered 77.6% of BRD4 interacting proteins (Supplementary Fig. 9b). Next, different illumination time points (2, 5, 10, 20 min) were used to regulate the labeling radius (Fig. 4a and

Supplementary Data 3). We reasoned that at shorter illumination time, PDPL would primarily label direct binding partners, while long times would include proteins identified in the shorter photoactivation period as well as label indirect targets in complexes. Indeed, we found a high overlap between adjacent time points (84.6% for 2 vs. 5 min; 87.7% for 5 vs. 10 min; 98.7% for 10 vs. 20 min) (Fig. 4b and Supplementary Fig. 9c). In all experimental groups, we not only detected BRD4 self-labeling, but also several of its known targets, such as MED1, CHD8, BICRA, NIPBL, SMC1A, and HMGB1 as annotated in the String database. The ion intensity of those targets is proportional to illumination time (Fig. 4c and Supplementary Fig. 9d). GO analysis of proteins identified in the 2 min group revealed the identified proteins are localized in the nucleus and participate in chromatin remodeling and RNA polymerase function. The molecular functions of the proteins were enriched in chromatin binding or transcription coactivation, consistent with the function of BRD4 (Fig. 4d). Protein interaction analysis enabled by the String database revealed the first layer of indirect interaction between BRD4 and HDAC family interacting complexes such as SIN3A, NCOR2, BCOR, and SAP130 (Fig. 4e and Supplementary Fig. 9e), in line with both BRD4 and HDACs binding acetylated histones. Furthermore, representative targets including Sin3A, NSUN2, Fus, and SFPQ identified by LC-MS/MS were validated by Western blot (Fig. 4f). Recently, a short isoform of BRD4 was reported to form nuclear puncta that possess liquid-liquid phase separation (LLPS) properties[40]. The RNA-binding proteins Fus and SFPQ, which mediate LLPS for a variety of cellular processes[41], were identified here as unreported BRD4 binding proteins. Co-immunoprecipitation (co-IP) experiments verified the

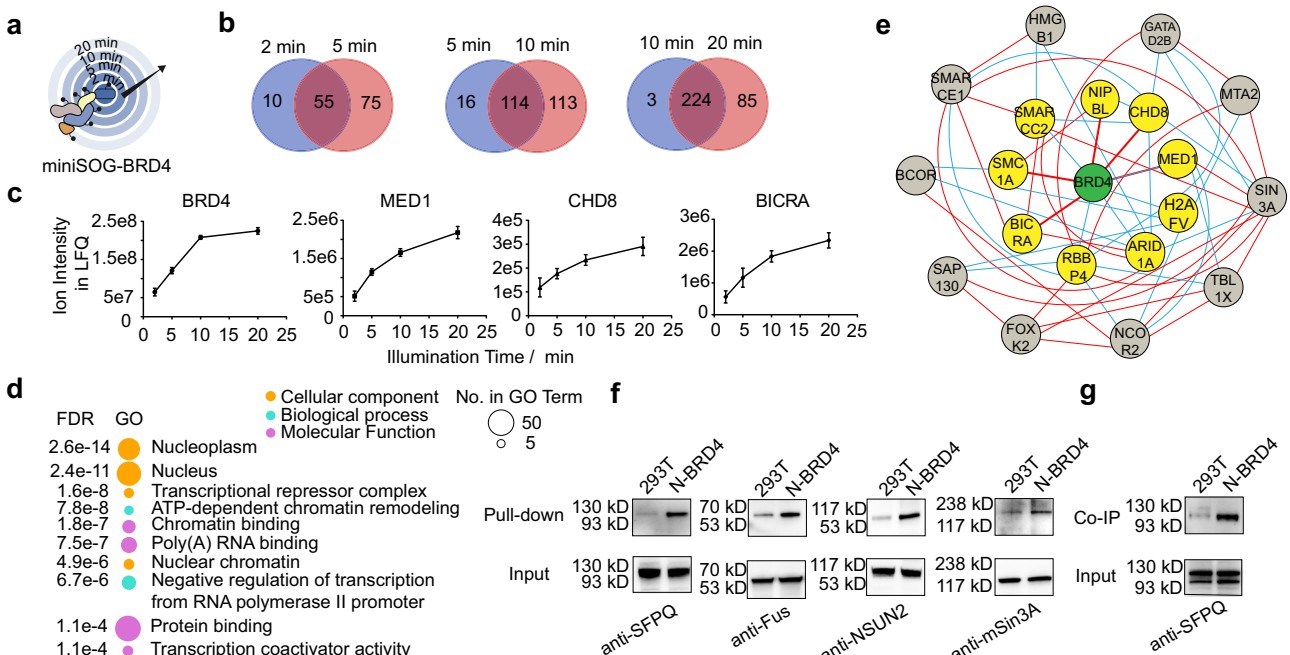

**Fig. 4 | Spatiotemporally-resolved profiling of BRD4 binding proteins with different labeling radii triggered by illumination duration. a** Schematic of miniSOG-mediated BRD4 proximity labeling with irradiation time: 2, 5, 10, and 20 min. **b** Overlapping of the identified proteins for different illumination time points. The enrichment of the identified proteins is statistically significant in HEK293T-miniSOG-BRD4 compared with wild-type HEK293T. **c** The ion intensity in label-free quantification for representative known BRD4 binding proteins across the indicated irradiation times. $n = 3$ biologically independent samples. Data are presented as mean values ± SD. **d** Gene ontology (GO) analysis of proteins identified in 2 min group. The top ten GO terms are listed. Bubbles are colored according to to GO term class and bubble size is proportional to the number of proteins found in each term. **e** String analysis of BRD4 interacting proteins. Yellow circles are direct binders and gray circles are in the first layer of indirect binders. The red lines indicate experimentally determined interactions and cyan lines indicate predicted interactions. **f** Representative BRD4 binding targets identified in LC-MS/MS were validated using Western blot. **g** Co-immunoprecipitation experiments verified the interaction between SFPQ and BRD4. These experiments were independently repeated at least twice with similar results. Source data are provided as a Source Data file.

interaction between BRD4 and SFPQ (Fig. 4g), indicating a different mechanism of BRD4 mediated liquid-liquid phase separation and warrant further investigation. Taken together, these results demonstrate PDPL to be an ideal platform to identify known BRD4 interactors as well as unreported binding proteins.

## Identification of Parkin substrates

In addition to identifying unreported binding targets of POI, we envisioned that PDPL would be suitable for identifying substrates of enzymes, which requires characterization of indirect binding proteins in a large complex to annotate unreported substrates. Parkin (encoded by PARK2) is an E3 ligase, and mutations in Parkin are known to cause autosomal recessive juvenile Parkinson's disease (AR-JP)[42]. Moreover, Parkin is described to be crucial for mitophagy (autophagy of mitochondria) and clearance of reactive oxygen species[43]. However, the function of Parkin in this disease is unclear despite the identification of several of its substrates. To annotate its uncharacterized substrates, PDPL was tested by incorporating miniSOG at the N or C termini of Parkin. Cells were treated with protonophore carbonyl cyanide m-chlorophenyl hydrazone (CCCP) to activate Parkin via the PINK1-Parkin pathway. In contrast to our BRD4 PDPL results, the Parkin N-terminal fusion identified a much larger set of protein targets, even though it covers most of the C-terminal (177 out of 210) (Fig. 5a, b and Supplementary Data 4). This result is in line with the report that N-terminal tagging can aberrantly activate Parkin[44]. Surprisingly, our data has only 18 overlapping proteins with published AP-MS results for Parkin[43], likely due to differences between cell lines and proteomics workflows. PDPL could exclusively identify 11 known binders of Parkin (ATXN2, IKBKG, PSMD4, TP53,

SUMO1, PSMD9, STUB1, PSMD4, DNAJB1, UBE2Z, and EPS15), in addition to four known proteins identified by both approaches (ARDM1, HSPA8, PSMD14, and PSMC3) (Fig. 5c)[43]. To further validate the LC-MS/MS results, PDPL processing and subsequent Western blot analysis were used to compare pull-down results for parental HEK293T cells and the N-terminal Parkin stable line. The previously unknown targets CDK2, DUT, CTBP1, and PSMC4 were validated together with the known binder DNAJB1 (Fig. 5d).

Notably, the proteins identified by PDPL should include binding proteins of Parkin as well as its substrates. To discover unreported substrates of Parkin, we selected seven identified proteins (PUF60, PSPC1, UCHL3, PPP1R8, CACYBP, Ssu72, and SNW1) and transfected plasmids, baring these genes into normal HEK293T as well as HEK293T stably expressing miniSOG-Parkin, followed by CCCP treatment. Ssu72 and SNW1 protein levels were significantly decreased in miniSOG-Parkin stable lines (Fig. 5e). CCCP treatment for 12 h gave the most significant degradation of both substrates. To investigate whether Ssu72 and SNW1 degradation were regulated by the ubiquitination-proteasome pathway, proteasome inhibitor MG132 was added to inhibit proteasome activity, and indeed, we found their degradation process was inhibited (Fig. 5f). The other non-substrate targets were validated as interactors of Parkin using Western blot (Supplementary Fig. 10), which showed consistent results with LC-MS/MS. Taken together, integration of the PDPL workflow with target protein transfection validation is able to identify unreported substrates for E3 ligases.

## Discussion

We have developed a general proximity labeling platform that permits spatiotemporally-resolved identification of interactors of POI. This

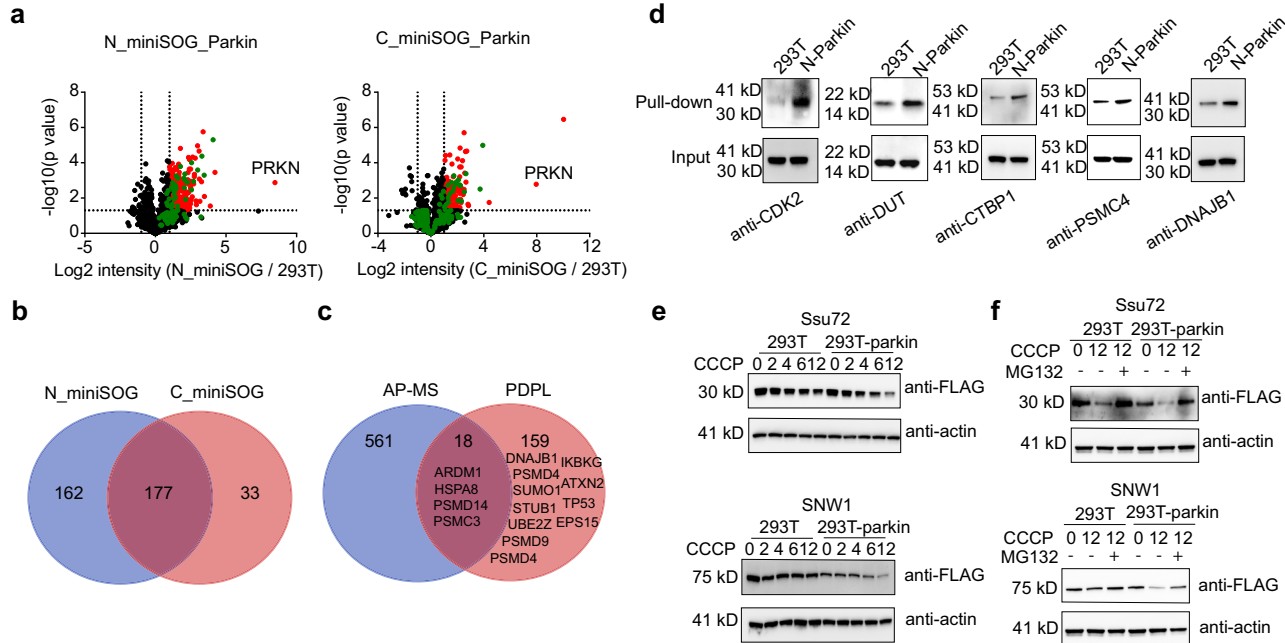

**Fig. 5 | Substrate identification of the E3 ligase Parkin. a** Volcano plots of Parkin-interacting proteins in HEK293T cells with stably expressed miniSOG fused to either the N-terminal or C-terminal of Parkin ($n = 3$ independent biological experiments). A two-sided student's $t$-test was used in the volcano plot. HEK293T was used as the negative control. Significantly changed proteins are highlighted in red ($p < 0.05$ and >2-fold ion intensity difference). Relevant proteins that are significant in HEK293T-miniSOG but not in HEK293T are marked in green. **b** Venn diagram showing overlapping proteins between N-terminal and C-terminal constructs. N-terminal tagging can aberrantly activate Parkin and lead to more identified proteins. **c** Venn diagram showing overlapping proteins between PDPL and AP-MS. Known interactors are listed, including four proteins out of 18 overlapped proteins, as well as 11 proteins out of 159 exclusively identified in PDPL. **d** Representative targets identified by LC-MS/MS were validated using Western blot. **e** Ssu72 and SNW1 were identified as unreported substrates of Parkin. Plasmids with these proteins labeled with FLAG were transfected into HEK293T and HEK293T-Parkin-miniSOG, followed by CCCP treatment across different time points. The degradation is more significant in the Parkin overexpressing line. **f** The Ssu72 and SNW1 degradation process was confirmed to be mediated by the ubiquitination-proteasome pathway by using the proteasome inhibitor MG132. These experiments were independently repeated at least twice with similar results. Source data are provided as a Source Data file.

platform is based on the photosensitizer protein miniSOG, which is only ~12 kD, less than half the size of the well-developed enzyme APEX2 (27 kD) and one-third the size of TurboID (35 kD). The smaller size should greatly expand the application scope for studying the interactome of small proteins. Future exploration of additional photosensitizers, both genetically-encoded proteins or small molecules[45], is warranted to increase the singlet oxygen quantum yield and expand the sensitivity of this approach. For the current miniSOG version, the use of blue light illumination to initiate proximity labeling, allows for high temporal resolution. Moreover, longer irradiation periods release a larger "cloud" of singlet oxygen, leading to the modification of more distal histidine residues, expanding the labeling radius and allowing for fine-tuning of the spatial resolution of PDPL. We also screened seven chemical probes to increase the signal-to-background ratio and explored the molecular mechanism underlying this approach. TOP-ABPP workflow coupled with unbiased open search confirmed that the modification occurred only at histidine, and no consistent micro-environment was observed for increased histidine modification, besides a modest preference for histidine at loop regions.

PDPL was also utilized to characterize subcellular proteomes, where the specificity and proteome coverage were at least comparable with other proximity labeling methods and organelle-specific chemical probe-based methods. Proximity labeling has also been successfully applied in characterizing surfacome, lysosome proteome, and secretory pathway-associated proteomes[46,47]. We believe PDPL would be compatible with these subcellular organelles. Furthermore, we challenged PDPL with the identification of cytosolic protein binding targets, which is more difficult than membrane-bound proteins due to their dynamic nature and involvement in more transient interactions.

PDPL was applied to two proteins, the transcriptional coactivator BRD4 and the disease-related E3 ligase Parkin. These two proteins were selected not only for their basic biological functions but also for their clinical relevance and therapeutic potential. Well-known binding partners, as well as unreported targets, were identified for both POIs. Notably, a phase separation-related protein SFPQ was validated by a co-IP, which may indicate a new mechanism in which BRD4 (short isoform) regulates LLPS. Meanwhile, the identification of Parkin substrates, we believe, is a scenario in which identification of indirect binders is required. We identified two unreported Parkin substrates and validated their degradation by the ubiquitination-proteosome pathway. Very recently, a mechanism-based trap strategy was developed to discover hydrolase substrates by capturing the substrates with the enzyme[48]. While an extremely powerful approach, it is not suitable for profiling substrates involved in the formation of large complexes and requires the formation of a covalent bond between enzymes and substrates. We anticipate that PDPL could be expanded to the study of other protein complexes and enzyme families, such as the deubiquitinase and metalloprotease families.

A new form of miniSOG, termed SOPP3, has been engineered with improved singlet oxygen yield[49]. We compared miniSOG to SOPP3 and found the labeling efficiency was increased, although the signal-to-noise remained unchanged (Supplementary Fig. 11). We posit that SOPP3 optimization (e.g., via directed evolution) would result in a more efficient photosensitizer protein requiring a much shorter illumination time, allowing for the capture of more dynamic cellular processes. Notably, the current version of PDPL is limited to the cellular environment as it requires blue light illumination, which is not deep-tissue penetrable. This feature prevents its utility in animal model

studies. However, optogenetics technologies coupled with PDPL could provide an avenue for animal study[50], especially in the brain. In addition, other engineered infrared photosensitizers would also alleviate this limitation. Research in this direction is currently in progress.

## Methods

### Cell culture
HEK293T cell line was obtained from ATCC (CRL-3216). The cell line has been tested negative for mycoplasma contamination and was cultured in DMEM (Thermo, #C11995500BT) supplemented with 10% fetal bovine serum (FBS, Vistech, #SE100-B) and 1% Penicillin/Streptomycin (Hyclone, #SV30010).

### Chemical probes
3-aminophenylene (probe **3**) and (4-ethynylphenyl)methanamine (probe **4**) were purchased from Bidepharm. Propyl amine (probe **2**) was purchased from Energy-chemicals. N-(2-aminophenyl)pent-4-ynamide (probe **1**) was synthesized according to a published procedure[51].

### Plasmid construction
Supplementary Table 1 lists the genetic constructs used in this study. The miniSOG and KillerRed sequences were cloned from gift plasmids from P. Zou (Peking University). Mitochondrial matrix targeting sequence was derived from the N-terminal 23-amino acids of *COX4*, then cloned into specified vectors by Gibson assembly (Beyotime, #D7010S). As for targeting to endoplasmic reticulum membrane and nucleus, human *SEC61B* (NM_006808.3) DNA amplified via PCR (NEB, #M0491L) from HEK293T cell cDNA library and *H2B* DNA (a gift from D. Lin at Shenzhen Bay Laboratory) were used and cloned as above. Other protein genes used in transfection and stable cell line construction were amplified via PCR from HEK293T cell cDNA library if not mentioned otherwise. G3S (GGGS) and G4S (GGGGS) were used as the linkers between the bait protein and miniSOG. A V5 epitope tag (GKPIPNPLLGLDST) was added to these fusion constructs. For mammalian expression and creation of stable cell lines, miniSOG fusion constructs were subcloned into a lentiviral vector pLX304. For bacterial expression, miniSOG was cloned into a pET21a vector with 6xHis-tag at the C-terminus.

### Stable cell line generation
HEK293T cells were seeded in a six-well plate at $2.0 \times 10^5$ cells per well and transfected after 24 h with a recombinant lentiviral plasmid (2.4 μg pLX304) and virus packaging plasmids (1.5 μg psPAX2 and 1.2 μg pMD2.G) using Lipo8000 (Beyotime, #C0533) at ~80% confluency. Following overnight transfection, media was exchanged and allowed to incubate for an additional 24 h. The viral collection was performed at 24, 48, and 72 h. Viral media was filtered with a 0.8 μm filter (Merck, #millex-GP) and Polybrene (Solarbio, #H8761) was added to a concentration of 8 μg/mL before infection of target cell lines. After 24 h, cells were allowed to recover by exchanging the media. Cells were selected with Blasticidin (Solarbio, #3513-03-9) at 5 μg/mL for the first three passages as a lower stringency selection. Then, 20 μg/mL was employed as a higher stringency for the following three passages.

### PDPL in living cells for fluorescence imaging analysis
Cells were seeded in a 12-well chamber (Ibidi, #81201) at a density of ~20,000 cells per well. To improve the adherence of HEK293T cells, chambers were pretreated with 50 μg/ml fibronectin (Corning, #356008) diluted in phosphate-buffered saline (PBS, Sangon, #B640435) for 1 h at 37 °C and removed by PBS. After 24 h, cells were washed with PBS once, incubated with 1 mM probe **3** in fresh Hanks Balanced Salt Solution (HBSS, Gibco, #14025092) for 1 h at 37 °C and then illuminated with blue LED (460 nm) for 10 min at room temperature. Thereafter, cells were washed with PBS twice and fixed with 4% formaldehyde (Sangon, #E672002) in PBS at room temperature for

15 min. Excess formaldehyde was removed from fixed cells through washing with PBS three times. Cells were then permeabilized with 0.5% Triton X-100 (Sangon, #A600198) in PBS and then washed three more times with PBS. Next, chamber was removed and 25 μL mixture of click reaction reagents was added to each sample, containing 50 μM Cy3-azide (Aladdin, #C196720), 2 mM CuSO₄ (Sangon, #A603008), 1 mM BTTAA (Confluore, #BDJ-4) and 0.5 mg/ml sodium ascorbate (Aladdin, #S105024), and incubated at room temperature for 30 min. After the click reaction, cells were washed with PBS containing 0.05% Tween-20 (Sangon, # A600560) (PBST) six times and then blocked with 5% BSA (Abcone, #B24726) in PBST for 30 min at room temperature.

For immunostaining to enable the colocalization analysis, cells were incubated with primary antibodies according to indicated conditions: anti-V5 tag mouse monoclonal antibody (1:500, CST, #80076), anti-Hsp60 rabbit monoclonal antibody (1:1000, ABclonal, #A0564), anti-calnexin rabbit polyclonal antibody (1:500, Abcam, #ab22595), or anti-Lamin A/C rabbit monoclonal antibody (1:500; CST, #2032) overnight at 4 °C. After washing with PBST three times, cells were incubated with secondary antibody: goat anti-rabbit Alexa Fluor 488 (Thermo, #A11034) at 1:1000 dilution, goat anti-mouse Alexa Fluor 594 (CST, #8889) at 1:1000 dilution for 30 min at room temperature. Cells were then washed three times with PBST and counterstained with DAPI (Thermo, #D1306) in PBS for 10 min at room temperature. Cells were sealed in 50% glycerol (Sangon, #A600232) in PBS for imaging after washing three times with PBS. Immunofluorescence images were collected with ZEISS LSM 900 Airyscan2 confocal microscope with software ZNE 3.5.

### Fluorescent imaging of singlet oxygen and mitochondrial superoxide
For fluorescent imaging of singlet oxygen, cells were washed with Hanks' HEPES buffer twice, then 100 nM Si-DMA (DOJINDO, #MT05) in Hanks' HEPES buffer was added. After illumination, the cells were cultured for 45 min in a CO₂ incubator at 37 °C. Thereafter, the cells were washed with Hanks' HEPES buffer twice and counterstained with Hoechst in Hanks' HEPES buffer for 10 min at room temperature and imaged with ZEISS LSM 900 confocal microscope. For fluorescent imaging of superoxide, 5 μM MitoSOX™ Red Mitochondrial Superoxide Indicator (Invitrogen, #M36008) in HBSS buffer containing calcium and magnesium was added to the cells. After illumination or Doxorubicin (MCE, #HY-15142A) treatment, the cells were cultured for 10 min in a CO₂ incubator at 37 °C, washed twice with HBSS buffer, and counterstained with Hoechst in HBSS buffer for 10 min at room temperature. Doxorubicin was used as the positive control for the probe where cells were treated with 20 μM Doxorubicin in HBSS containing 1% BSA for 30 min. Immunofluorescence images were collected with ZEISS LSM 900 confocal microscope.

### Identification of labeling sites of PDPL
HEK293T cells stably expressing mito-miniSOG were seeded in 15-cm dishes at a density of ~30%. After 48 h, when reached ~80% confluency, cells were washed with PBS once, incubated with 1 mM probe **3** in fresh HBSS buffer for 1 h at 37 °C, and then illuminated with blue LED for 10 min at room temperature. Thereafter, cells were washed with PBS twice, scraped, and resuspended in an ice-cold PBS buffer containing EDTA-free protease inhibitor (MCE, # HY-K0011). The cells were lysed by tip sonication for 1 min (1 s on and 1 s off, 35% amplitude). The resulting mixture was centrifuged at 15,871×g for 10 min at 4 °C to remove the debris and the concentration of the supernatant was adjusted to 4 mg/mL using a BCA protein assay kit (Beyotime, #P0009). 1 mL of the above lysate was incubated with 0.1 mM photocleavable biotin-azide (Confluore, #BBBD-14), 1 mM TCEP (Sangon, #A600974), 0.1 mM TBTA (Aladdin, #T162437) ligand, and 1 mM CuSO₄ for 1 h with bottom-up rotation at room temperature. After click reaction, the mixture was added to a pre-mixed solution (MeOH:

$CHCl_3$: $H_2O$ = 4 mL: 1 mL: 3 mL) in a 10 mL glass bottle. Samples were mixed and centrifuged at 4500×$g$ for 10 min at room temperature. The bottom and upper layer solution was discarded sequentially, and the pellet was subsequently washed twice with 1 mL methanol followed by centrifuging at 15,871×$g$ for 5 min at 4 °C. 1 mL of 8 M urea (Aladdin, #U111902) in 25 mM ammonium bicarbonate (ABC, Aladdin, #A110539) was added to dissolve the pellet. The samples were reduced with 10 mM dithiothreitol (Sangon, #A100281, in 25 mM ABC) at 55 °C for 40 min and then alkylated by adding 15 mM fresh iodoacetamide (Sangon, #A600539) in dark at room temperature for 30 min. An additional 5 mM of dithiothreitol was added to stop the reaction. About 100 µL NeutrAvidin agarose resin beads (Thermo, #29202) for each sample were prepared by washing three times with 1 mL PBS. The above proteome solution was diluted with 5 mL PBS and incubated with pre-washed NeutrAvidin agarose resin beads for 4 h at room temperature. Next, the beads were washed with 5 mL PBS containing 0.2% SDS (Sangon, #A600485) three times, 5 mL PBS containing 1 M urea three times, and 5 mL ddH$_2$O three times. The beads were then collected by centrifugation and resuspended in 200 µL 25 mM ABC containing 1 M urea, 1 mM CaCl$_2$ (Macklin, #C805228), and 20 ng/µL trypsin (Promega, #V5280). Trypsin digestion was performed at 37 °C with rotation overnight. The reaction was quenched by adding formic acid (Thermo, # A117-50) till pH reached 2–3. The beads were washed with 1 ml PBS containing 0.2% SDS three times, 1 ml PBS containing 1 M urea three times, and then 1 ml distilled water three times. Release of modified peptides by photo (365 nm) cleavage for 90 min with 200 µL 70% MeOH. After centrifugation, the supernatant was collected. Then the beads were washed once with 100 µL of 70% MeOH and supernatant was combined. The samples were dried in a speedvac vacuum concentrator and stored at −20 °C until analysis.

For the identification and quantification of singlet oxygen-modified peptides, the samples were redissolved in 0.1% formic acid and 1 µg peptides were analyzed with an Orbitrap Fusion Lumos Tribrid mass spectrometer equipped with a nano-ESI source with the vendor-provided Tune and Xcalibur 4.3 software. The samples were separated on an in-house packed 75 µm × 15 cm capillary column with 3 µm C18 material (ReproSil-pur, #r13.b9.) and connected to an EASY-nLC 1200 UHPLC system (Thermo). Peptides were chromatographically separated by a linear 95 min gradient from 8 to 50% solvent B (A = 0.1% formic acid in water, B = 0.1% formic acid in 80% acetonitrile) and followed by a linear increase to 98% B in 6 min at a flow rate of 300 nL/min. The Orbitrap Fusion Lumos acquired data in a data-dependent manner alternating between full-scan MS and MS2 scans. The spray voltage was set at 2.1 kV and the temperature of the ion transfer capillary was 320 °C. The MS spectra (350 – 2000 m/z) were collected with 120,000 resolution, AGC of $4 \times 10^5$, and 150 ms maximal injection time. The top ten most abundant multiply charged precursors from each full scan were fragmented by HCD with 30% normalized collision energy, quadrupole isolation windows of 1.6 m/z, and a resolution setting of 30,000. AGC targets for tandem mass spectrum of $5 \times 10^4$ and 150 ms maximal injection time were used. Dynamic exclusion was set to 30 s. Unassigned ions or those with a charge of 1+ and >7+ were rejected for MS/MS.

The raw data were processed using the MSFragger-based FragPipe computational platform[32]. Open search algorithm with precursor mass tolerance −150 to 500 Da were used to determine the mass shift and corresponding amino acids. Then modifications on histidine with delta mass +229.0964 and +247.1069 Da were used in PD (proteome discoverer 2.5, Thermo) to identify the modified peptides.

### PDPL in living cells for gel analysis

Cells stably expressing miniSOG fusion genes were seeded in a 6-cm dish. When reached ~80% confluency, cells were washed once with HBSS (Gibco, #14025092), followed by incubation with chemical probes in HBSS at 37 °C for 1 h and illumination with a 10 W blue LED

for 20 min at room temperature. To determine which type of reactive oxygen species is involved in PDPL, 0.5 mM Vitamin C (MCE, #HY-B0166), 5 mM Trolox (MCE, #HY- 101445), D$_2$O (Sigma, #7789-20-0), 100 mM Mannitol (Energy Chemical, #69-65-8), 100 µM H$_2$O$_2$, 10 mM NaN$_3$ were added to the cells as the additives. Following rinse with cold PBS, cells were scraped, collected into a 1.5 mL centrifuge tube, and lysed in 200 µL PBS with 1x EDTA-free protease inhibitor using tip sonication for 1 min (1 s on and 1 s off, 35% amplitude). The resulting mixture was centrifuged at 15,871×$g$ for 10 min at 4 °C and the concentration of the supernatant was adjusted to 1 mg/mL using a BCA protein assay kit. About 50 µL of the above lysate was incubated with 0.1 mM rhodamine-azide (Aladdin, #T131368), 1 mM TCEP, 0.1 mM TBTA ligand, and 1 mM CuSO$_4$ for 1 h with bottom-up rotation at room temperature. After click reaction, acetone precipitation was performed by adding 250 µL pre-cooled acetone to the sample, incubation at −20 °C for 20 min, and centrifuging at 6010×$g$ for 10 min at 4 °C. The pellet was collected and boiled in 50 µL 1x Laemmli buffer for 10 min at 95 °C. Samples were then run in SDS-PAGE long gel, visualized by Bio-rad ChemiDoc MP Touch imaging system with Image Lab Touch Software.

### In vitro labeling

miniSOG-6xHis recombinant protein expression and purification were performed as described previously[18]. Briefly, *Escherichia coli* BL21(DE3) (TransGen, #CD701-02) cells were transformed with pET21a-miniSOG-6xHis and induced protein expression by 0.5 mM IPTG (Sangon, #A600168). After cell lysis, the protein was purified via Ni-NTA agarose beads (MCE, #70666), dialyzed against PBS, and stored at – 80 °C.

For antibody-based proximity labeling assay in vitro, 100 µM purified miniSOG, 1 mM probe **3** and 1 µg anti-his-tag mouse monoclonal antibody (TransGen, #HT501-01) were mixed in PBS with 50 µL total reaction volume. The reaction mixture was irradiated with blue LED light for 0, 2, 5, 10, and 20 min at room temperature. The mixture was incubated with 0.1 mM biotin-PEG3-azide (Aladdin, #B122225), 1 mM TCEP, 0.1 mM TBTA ligand, and 1 mM CuSO$_4$ on a bottom-up shaker at room temperature for 1 h. After click reaction, 4x Laemmli buffer was added to the mixture directly and boiled for 10 min at 95 °C. The samples were run on SDS-PAGE gel and analyzed by streptavidin-HRP (1:1000, Solarbio, #SE068) western blot.

For peptide-based proximity labeling assay in vitro, a histidine-containing synthetic peptide (LHDALDAK-CONH2) with C-terminal amidation was used. In this assay, 100 µM purified miniSOG, 10 mM probe **3**, and 2 µg/mL synthetic peptide were mixed in PBS with a 50 µL total reaction volume. The reaction mixture was irradiated with blue LED light for 1 h at room temperature. One microliter sample was analyzed by LC-MS system (Waters, SYNAPT XS Ions Mobility Time-of-Flight Mass Spectrometer with MassLynx Spectrum analysis software).

### PDPL in living cells for LC-MS/MS analysis

HEK293T cells stably expressing miniSOG fusion genes were seeded in 10-cm dishes for different organelle localization lines (Mito, ER, Nucleus) and in 15-cm dishes for Parkin-miniSOG and BRD4-miniSOG lines. When reached to ~90% confluency, cells were washed once with HBSS, followed by incubation with probe **3** in HBSS at 37 °C for 1 h and illumination with a 10 W blue LED as indicated at room temperature. For Parkin proximity labeling, 10 µM protonophore carbonyl cyanide m-chlorophenyl hydrazone CCCP (Solarbio, #C6700) was added together with probe **3** in HBSS at 37 °C for 1 h. Cell lysis, click chemistry, reduction, and alkylation steps were the same as above, except 2 mg lysate was input and biotin-PEG3-azide instead of photo-cleavable biotin-azide was used in click reaction. After beads enrichment, the beads were washed with 5 mL PBS containing 0.2% SDS three times, 5 mL PBS containing 1 M urea three times, and 5 mL PBS three times. After that, 2 µg of trypsin in 300 µL of 25 mM ABC containing 1 M urea was added for protein digestion overnight at 37 °C. The reaction was

quenched by adding formic acid till pH reached 2–3. After on-bead trypsin digestion, the peptide solution was desalted using the SOLAµ HRP column (Thermo, #60209-001) and dried in speedvac vacuum concentrator. Peptides were redissolved in 0.1% formic acid and 500 ng peptides were analyzed with an Orbitrap Fusion Lumos Tribrid mass spectrometer equipped with a nano-ESI source as described above. The peptides were separated on a commercial RP-HPLC pre-column (75 µm × 2 cm) (Thermo, #164946) and RP-HPLC analytical column (75 µm × 25 cm) (Thermo, #164941), both packed with 2 µm C18 beads using a linear gradient ranging from 8 to 35% ACN in 60 min and followed by a linear increase to 98% B in 6 min at a flow rate of 300 nL/min. The MS spectra (350 − 1500 m/z) were collected with 60,000 resolution, AGC of $4 \times 10^5$, and 50 ms maximal injection time. Selected ions were sequentially fragmented in a 3 s cycle by HCD with 30% normalized collision energy, quadrupole isolation windows of 1.6 m/z, and 15,000 resolution. AGC targets for tandem mass spectrum of $5 \times 10^4$ and 22 ms maximal injection time were used. Dynamic exclusion was set to 45 s. Unassigned ions or those with a charge of 1+ and >7+ were rejected for MS/MS.

### PDPL in living cells for western blot analysis

The sample preparation steps till NeutrAvidin beads enrichment were the same as in LC-MS/MS analysis mentioned above. About 50 µg lysate was used as the loading control input and 2 mg lysate was used for click reaction. After NeutrAvidin enrichment and washing, the binding proteins were eluted by adding 50 µL Laemmli buffer to the agarose resin beads and boiling for 5 min at 95 °C. Loading control input and beads enriched samples were analyzed by SDS-PAGE and transferred to PVDF membranes (Millipore, #ISEQ00010) by standard western blotting methods. Membranes were blocked in 5% non-fat milk (San-gon, #A600669) in TBS containing 0.1% tween-20 (TBST) and incu-bated with primary and secondary antibodies sequentially. Primary antibody was used at 1:1000 dilution in 5% non-fat milk in TBST and incubated overnight at 4 °C. Secondary antibodies were used at 1:5000 and incubated for 1 h at room temperature. The membranes were visualized using chemiluminescence by the Chemidoc MP imaging system. Uncropped scans of all blots and gels in Figures are supplied as Source Data.

Primary antibodies used in this study include anti-SFPQ rabbit monoclonal antibody (CST, #71992), anti-FUS rabbit monoclonal antibody (CST, #67840), anti-NSUN2 rabbit polyclonal antibody (Pro-teintech, #20854-1-AP), anti-mSin3A rabbit polyclonal antibody (Abcam, #ab3479), anti-Flag tag mouse monoclonal antibody (Trans-Gen, #HT201-02), anti-β-actin mouse monoclonal antibody (TransGen, #HC201-01), anti-CDK2 rabbit monoclonal antibody (ABclonal, #A0094), anti-CTBP1 rabbit monoclonal antibody (ABclonal, #A11600), anti-DUT rabbit polyclonal antibody (ABclonal, #A2901), anti-PSMC4 rabbit polyclonal antibody (ABclonal, #A2505), anti-DNAJB1 rabbit polyclonal antibody (ABclonal, #A5504). These anti-bodies were used at 1:1000 dilution in 5% non-fat milk in TBST. Sec-ondary antibodies used in this study include anti-rabbit IgG (TransGen, #HS101-01), anti-mouse IgG (TransGen, #HS201-01) at 1:5000 dilution.

### Co-immunoprecipitation

To further examine whether BRD4 interacts with SFPQ, HEK293T and BRD4-miniSOG overexpressing HEK293T stable cells were seeded in 10-cm dishes. Cells were washed with cold PBS and lysed in 1 ml Pierce IP Lysis Buffer (Thermo Fisher, #87787) with 1x EDTA-free protease inhibitor for 30 min at 4 °C. Thereafter, lysates were col-lected in 1.5 mL centrifuge tube and centrifuged at 15,871×g for 10 min at 4 °C. The supernatants were collected and incubated with 5 µg anti-V5 tag mouse monoclonal antibody (CST, #80076) over-night at 4 °C. About 50 µL protein A/G magnetic beads (MCE, #HY-K0202) were washed twice with PBS containing 0.5% tween-20. The cell lysate was subsequently incubated with the magnetic beads with

bottom-up rotation for 4 h at 4 °C. Then, the beads were washed four times with 1 mL PBST buffer and boiled for 5 min at 95 °C. Samples were run in SDS-PAGE gel and transferred to PVDF membranes by standard western blotting methods. Membranes were blocked in 5% non-fat milk in TBST and incubated with primary and secondary antibodies sequentially. Primary antibody anti-SFPQ rabbit mono-clonal antibody (CST, #71992) was used at 1:1000 dilution in 5% non-fat milk in TBST and incubated overnight at 4 °C. anti-rabbit IgG was used at 1:5000 and incubated for 1 h at room temperature. The membranes were visualized using chemiluminescence by the Che-midoc MP imaging system.

### Solvent-accessible surface area (SASA) analysis

All the structures used for solvent-accessible surface area (SASA) analysis were obtained from the Protein Data Bank (PDB)[52] or the AlphaFold Protein Structure Database[53]. The absolute SASA of each residue was computed using the FreeSASA program[54]. Only SASA data of both the labeled histidines and their neighbors that were complete and unambiguous were used to obtain the average SASA for each structure. The Relative Solvent Accessibility (RSA) for each histidine was calculated by dividing absolute SASA values by the empirical maximum possible solvent-accessible surface area of the residue[55]. Then all the histidines were classified as buried if the average RSA were lower than 20% and exposed otherwise[56].

### Data analysis

Raw files acquired in DDA mode were searched against corresponding SwissProt-reviewed protein databases containing common con-taminants using Proteome Discoverer (v2.5) or MSfragger (Fragpipe v15.0). Peptides were required to be fully tryptic with a maximum of two missed cleavage sites, carbamidomethylation as fixed modifica-tion, and methionine oxidation as a dynamic modification. The pre-cursor and fragment mass tolerance were set to 10 ppm and 0.02 Da (MS2 orbitrap), respectively. Contaminant hits were removed, and proteins were filtered to obtain a false discovery rate of <1%. Normal-ized protein abundances from three biological replicates were used for label-free quantification analysis. Protein subcellular localization ana-lysis was enabled by Gene Ontology (GO) analysis from DAVID Bioin-formatics Resources, MitoCarta 3.0, and a database collected and published by Alice Ting group[8]. Volcano plots were acquired from Perseus (v1.6.15.0). Protein abundance fold changes were tested for statistical significance using a two-sided t-test, and protein hits were identified with abundance change >2 (unless otherwise stated) and p value <0.05. Protein interaction analysis was performed using GO analysis as well as the String database.

### Statistics and reproducibility

Three biological replicates were performed with similar results. Sta-tistical analysis was performed on GraphPad Prism (GraphPad Soft-ware) and volcano plots were acquired from Perseus (v1.6.15.0). For comparison between two groups, p values were determined using a two-sided Student's t-test. Singleton proteins which were only identi-fied in experimental groups at least twice, were included in the volcano plots, and the corresponding missing values in control groups were replaced from normal distribution by Perseus to enable the p value calculation. Error bars represent means ± SD. In proteomics analysis, the protein abundances that appear in at least two biological replicates were kept to enable the statistical analysis. No statistical method was used to predetermine sample size. The experiments were not rando-mized. The Investigators were not blinded to allocation during experiments and outcome assessment.

### Reporting summary

Further information on research design is available in the Nature Research Reporting Summary linked to this article.

## Data availability

The mass spectrometry data generated in this study have been deposited to the ProteomeXchange Consortium via the iProX[57] partner repository with the dataset identifier PXD034811 (PDPL-MS dataset). Source data are provided as a Source Data file. Source data are provided with this paper.

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

## Acknowledgements

We thank the mass spectrometry core facility, imaging core facility, and sequencing core facility in Shenzhen Bay Laboratory for their assistance in running samples. We thank Dr. Yu-Hsan Tsai, Dr. Xiaoyu Li, and Dr. Jeff Montgomery for helpful discussion and proofreading assistance. We are grateful for the financial support of this work from the following: Grant from the National Natural Science Foundation (32101200 to G.L.), Grant from Guangdong-Shenzhen Regional Joint Fund (2020A1515110903 to G.L.), and Grant from Shenzhen Bay Laboratory Open Fund (SZBL2020090501008 to G.L.).

## Author contributions

All authors reviewed the manuscript. G.L. conceived of the study and supervised research. Y.Z., X.H., K.Z., Y.H., and G.L. designed and analyzed experiments. Y.Z., X.H., K.Z., Y.H., J.C., W.Z., and C.K.C.C. performed experiments. Y.J. proposed the chemical mechanism. Z.Z. performed the SASA analysis. Y.Z. and G.L. wrote the manuscript.

## Competing interests

The authors declare no competing interests.
