## [Peer Review File · Nature Communications]

REVIEWER COMMENTS

Reviewer #1 (Remarks to the Author):

This manuscript by Zhai et al. describes a new method for proximity labeling called PDPL based on photoactivation with a miniSOG to generate activation of a reactive probe that labels proteins on histidine. This method was demonstrated to function in multiple cellular compartments, identified proteins in those compartments by protein MS. PDPL was also applied to two specific proteins, BRD4 in the nucleus and Parkin in the cytoplasm and some of the expected protein interactors of these proteins were identified and validated by co-IP. Overall the manuscript is properly organized, interpretable and provides information on a new method and shows evidence that it works.

Specific comments:

The authors should consider a more discretely located PDPL bait, for example nuclear pores, nuclear lamins, centrosomes, tight junctions, etc. This has been done with the other common proximity labeling methods to help show specificity of labeling. Without this, none of the data provided clearly demonstrate the labeling area of PDPL. Excessive labeling areas has been an issue with other similar methods, including those that claim to be very proximate with no evidence backing them up other than in vitro derived chemical half-lives of reactive products.

Fig S3A: There should be some kind of control to ensure that expression of these fusion proteins is similar. The actin loading control is good for total protein loading but not sufficient. If need be, some kind of shared tag between the SOG and TurboID proteins should be used to show if similar level of protein are being expressed. Plus, it is unusual that there are very different patterns of labeling by WB between the two methods. Could the authors show the IF images to confirm that these proteins were indeed in similar locations.

Fig 2F would be clearer and more accurate if the blot on the left were labeled anti-MAb, or anti-mouse or similar. Anti-His makes it seem like it has been probed with an anti-His antibody.

The histone H2B fusion to PDPL was an unusual choice since H2B is notoriously immobile, essentially fixed onto the chromatin. This is unlike the ER and Mito PDPLs that are more soluble/mobile within their compartments. It is perhaps unexpected that there would be such strong nucleolar labeling. What does this unexpected labeling in distal areas of the nucleus, away from the chromatin, say about the spatial aspects of PDPL?

For the BRD4-PDPL experiments, why not also compare to the H2B-PDPL results which might be a better control for specificity?

Would PDPL work in the secretory pathway, in lysosomes, or on the cell surface? Perhaps just discuss.

Reviewer #2 (Remarks to the Author):

In the manuscript "Spatiotemporal-resolved protein networks profiling with photoactivation dependent proximity labeling", Yansheng Zhai and his colleagues engineered a photoactivated platform for characterization of protein interaction based on spatial proximity. This innovative technology has great potential in biological science.

The work is performed thoroughly, experiments are proved carefully.

I would like to discuss the following concern.

The authors state that the protein labeling reaction is based on singlet oxygen generated by miniSOG. But miniSOG is really poor singlet oxygen sensitizer ($\Delta\Phi = 0.030 \pm 0.002$) [Pimenta, F. M.; Jensen, R. L.; Breitenbach, T.; Etzerodt, M.; Ogilby, P. R. *Oxygen-Dependent Photochemistry and Photophysics of "MiniSOG," a Protein-Encased Flavin*. *Photochem. Photobiol.* 2013, 89, 1116–1126; Ruiz-González, R.; Cortajarena, A. L.; Mejias, S. H.; Agut, M.; Nonell, S.; Flors, C. *Singlet Oxygen Generation by the Genetically Encoded Tag miniSOG*. *J. Am. Chem. Soc.* 2013, 135, 9564–9567]. It is well documented that FMN encapsulated in protein scaffold produces a very limited singlet oxygen amount due to the quenching of a significant fraction of the nascent 1O_2 molecules by aromatic amino acid residues of the protein. The new form of miniSOG, SOPP3 [M. Westberg et al. *No Photon Wasted: An Efficient and Selective Singlet Oxygen Photosensitizing Protein*. *J. Phys. Chem. B* 2017, 121, 9366–9371] has been created which generate singlet oxygen with yield 0.6. But the authors didn't use this form during the experiments.

The only evidence that the protein labeling reaction is mediated by singlet oxygen is given in fig.1e, showing the fluorescence in mitochondria in the presence of Si-DMA probe. First, I think, this experiment lacks a negative control: HEK293 cells without miniSOG expression treated with Si-DMA. And second, what about cells treated with superoxide anion sensor (for example, MitoSOX™ Red Mitochondrial Superoxide Indicator)?

Further, the authors state that "protein labeling was greatly reduced in the presence of sodium azide, which is known to quench singlet oxygen" (p. 6, fig. d). Of course, NaN_3 an efficient quencher of 1O_2 . However, caution should be exercised with this interpretation since NaN_3 is also a very efficient quencher of the FMN triplet state [Spikes, J. D., et al. *Photodynamic crosslinking of proteins. III. Kinetics*

of the FMN- and rose bengal-sensitized photooxidation and intermolecular crosslinking of model tyrosine-containing N-(2-hydroxypropyl)methacrylamide copolymers. *Photochem. Photobiol.* 1999, 70, 130–137].

As soon as miniSOG is very poor singlet oxygen sensitizer, it is worth to prove that the mechanism of proposed PDLP is indeed through singlet oxygen.

The manuscript can be published after major revision.

Reviewer #3 (Remarks to the Author):

Li and colleagues developed a novel proximity labeling method for capturing protein-protein interactions in a temporal manner from the complexed proteome samples. They fused a small photosensitizer protein, miniSOG, to the protein of interest (POI) and shed it with blue light to trigger the generation of singlet oxygen. They validated that the irradiation will oxidize histidine residues in nearby proteins which can then be labeled by an aniline-based chemical probe. The proximity-labeled proteomes can be analyzed by chemical proteomics to identify subcellular proteome composition as well as enzyme substrates. They applied the method to identify novel protein substrates of BRD4 and Parkin, and biochemically validated some of these targets. Overall, this is a well-conducted study that adds a new tool for proximity labeling other than APEX and TurboID. I support the publication of this study with some minor comments to be addressed by the authors.

- 1) The aniline probe has also been shown to label protein carbonylations in proteomes (<https://www.ncbi.nlm.nih.gov/pubmed/29569437>). Have the authors observed any labeling adducts to such modifications during their proteomic analysis?
- 2) Can they comment on how many histidines on average per protein are identified by the PDPL approach? Are these labeled histidines well exposed, e.g, in terms of SASA?
- 3) Most of the main figures are formatted with very small and over-crowding fonts, and should be reformatted for better clarity.
- 4) Figure 1a needs to be re-drawn. The amine warhead is drawn in a quite weird way. It did not convey the message that the singlet oxygen oxidizes proteins and the oxidized histine is captured then by the probe.
- 5) Texts need more careful proofreading as there are several grammatic errors and poorly worded sentences throughout the manuscript.

Point-by-point response to referee comments for manuscript #NCOMMS-22-11388

We would like to thank all three Reviewers for their time and insightful comments. Included with our revised manuscript is a marked version, where changes in the text are red color highlighted. We have made every attempt to address all concerns raised during review, and we believe our revised manuscript has been significantly improved in the process.

REVIEWER COMMENTS

Reviewer #1 (Remarks to the Author):

This manuscript by Zhai et al. describes a new method for proximity labeling called PDPL based on photoactivation with a miniSOG to generate activation of a reactive probe that labels proteins on histidine. This method was demonstrated to function in multiple cellular compartments, identified proteins in those compartments by protein MS. PDPL was also applied to two specific proteins, BRD4 in the nucleus and Parkin in the cytoplasm and some of the expected protein interactors of these proteins were identified and validated by co-IP. Overall the manuscript is properly organized, interpretable and provides information on a new method and shows evidence that it works.

We thank this Reviewer for their input on our work.

Specific comments:

The authors should consider a more discretely located PDPL bait, for example nuclear pores, nuclear lamins, centrosomes, tight junctions, etc. This has been done with the other common proximity labeling methods to help show specificity of labeling. Without this, none of the data provided clearly demonstrate the labeling area of PDPL. Excessive labeling areas has been an issue with other similar methods, including those that claim to be very proximate with no evidence backing them up other than in vitro derived chemical half-lives of reactive products.

We agree with the Reviewer that excessive labeling has been a common issue for proximity labeling method and the labeling radius was derived based on chemical half-lives of reactive products. Following this suggestion, we applied PDPL in nuclear lamins as a more discretely located example. As shown in figure S8, confocal microscopy study first verified the correct localization of miniSOG-Lamin A construct. PDPL workflow identified 36 significant enriched proteins, with 12 proteins (30.0%) being well-characterized Lamin A interacting proteins, representing a higher percentage than BioID method (28 out of 122 proteins, 22.9%) (Burke, B. et al. *J. Cell Biol.* **2012**, *6*, 801-810). Our method identified less proteins likely due to the restricted labeling area, enabled by more reactive singlet oxygen. GO analysis reveals that the identified proteins are mainly located in nucleoplasm (26), nuclear envelop (10), nuclear membrane (9), and nuclear pore (5). Combined, these nucleus-located proteins account for 80% of the enriched proteins, further demonstrating the specificity of PDPL.

Line 239-248: To define the labeling specificity of PDPL, nuclear Lamin A was selected as a more discretely localized POI bait⁷. PDPL identified 36 significantly enriched proteins, with 12 proteins (30.0%, including Lamin A) being well-characterized Lamin A interacting proteins annotated by String database, representing a higher percentage than BioID method (28 out of 122 proteins, 22.9%)⁷. Our method identified less proteins likely due to the restricted labeling area, enabled by more reactive singlet oxygen. GO analysis reveals that the identified proteins are mainly located in nucleoplasm (26), nuclear envelope (10), nuclear membrane (9), and nuclear pore (5). Combined, these nucleus-localized proteins account for 80% of the enriched proteins, further demonstrating the specificity of PDPL (Supplementary Fig. 8a-d).

Supplementary Figure 8 | a) Confocal imaging of HEK293T cells expressing miniSOG-Lamin A which were detected with Lamin A antibody, V5 tag antibody and DAPI. Scale bar: 10 μm. **b)** Volcano plot of PDPL-labeled proteome in miniSOG-Lamin A expressing cells by label free quantification (n = 3 independent biological experiments). Significantly changed proteins are highlighted in red ($p < 0.05$ and >1.5 -fold ion intensity difference). Relevant proteins that are significant in HEK293T-miniSOG but not in HEK293T are marked in green. **c)** GO analysis of cellular component revealed the subnuclear localization of the enriched proteins. p values were labeled on each bar plot. **d)** Specificity analysis for proteomic data set derived from experiments in **b**. Total number of statistically significant proteins was labeled on top. Bar plot shows the combined protein number in **c**. These experiments were independently repeated twice with similar results.

Fig S3A: There should be some kind of control to ensure that expression of these fusion proteins is similar. The actin loading control is good for total protein loading but not sufficient. If need be, some kind of shared tag between the SOG and TurboID proteins should be used to show if similar

level of protein are being expressed. Plus, it is unusual that there are very different patterns of labeling by WB between the two methods. Could the authors show the IF images to confirm that these proteins were indeed in similar locations.

We thank the Reviewer for these two suggestions. As shown in figure S6, anti-V5 tag Western blots were added to show that similar levels of proteins are being expressed. For nucleus-localization constructs, the expression of miniSOG and TurboID were similar; as for mitochondrion localization, TurboID was more expressed in contrast to the ER localization. Notably, we observed upshifted bands for miniSOG groups, where the intensities of the bands were reduced after amine probe **3** treatment. We posit that the upshifted bands are protein crosslinking products enabled by proximal lysine with oxidized histidine, and these products were competed and reduced by aniline probe. IF images of the TurboID constructed were added to show the correct locations for three organelles.

Supplementary Figure 6 | Side-by-side comparison of PDPL to TurboID in sub-organelle protein profiling. **a-c**) Representative gel imaging of three organelle-specific labeling comparisons of PDPL to TurboID. miniSOG and TurboID were located in mitochondria, nucleus and ER. 50 μ M and 500 μ M exogenous biotin were added in TurboID experiments. β -actin was used as the protein loading control and V5 tag was used to show similar levels of miniSOG and TurboID are being expressed. **d**) Fluorescent imaging of the TurboID constructs to show the correct localizations for three organelles. Scale bar: 10 μ m.

Fig 2F would be clearer and more accurate if the blot on the left were labeled anti-MAb, or anti-mouse or similar. Anti-His makes it seem like it has been probed with an anti-His antibody.

We thank the Reviewer for this comment. We modified the labels as “anti-mAb” instead of “anti-his” to be more accurate and clearer.

The histone H2B fusion to PDPL was an unusual choice since H2B is notoriously immobile, essentially fixed onto the chromatin. This is unlike the ER and Mito PDPLs that are more soluble/mobile within their compartments. It is perhaps unexpected that there would be such strong nucleolar labeling. What does this unexpected labeling in distal areas of the nucleus, away from the chromatin, say about the spatial aspects of PDPL?

We thank the Reviewer for this comment. Indeed, H2B would be considered as chromatin localized intuitively, and several papers utilized H2B for the proximity RNA labeling and DNA labeling (Zou, P. et al, *Nat. Chem. Biol.*, **2019**, *15*, 1110-1119; Zou, P. et al, *Angew. Chem. Int. Ed.*, **2020**, *59*, 22933-22937). However, fluorescent imaging of H2B dynamic assembly reveals that histone H2B initially accumulates in the nucleolus after expression and then gradually incorporates into the chromatin to leave a small amount of nucleolus-bound histone (Sheval, E. et al. *Biochim Biophys Acta Mol Cell Res BBA-MOL CELL RES*, **2011**, *1813*, 27-38.). Moreover, the nucleolus is proved to involve in H2B degradation (Chen, S. et al, *iScience*, **2021**, *24*, 102256). Therefore, these results may explain the nucleolus labeling by miniSOG-H2B.

To further confirm the nucleus labeling specificity, we constructed the miniSOG-3xNLS (nucleus localization signal) plasmid as an alternative and performed PDPL processing. As shown in figure S7, confocal imaging validated the nucleus localization of miniSOG. In-gel fluorescence analysis revealed the nucleus proteins labeling, and LFQ identified 592 nucleus proteins out of 830 significantly enriched proteins, accounting for 71.3% accuracy. The overlap between H2B and 3xNLS exceeded 67%, which demonstrated the accuracy of both H2B and 3xNLS methods.

Line 238-239: Nucleus proteome profiling by nuclear localization signals (3xNLS) peptide revealed similar accuracy to H2B construct (Supplementary Fig. 7c-h).

Supplementary Figure 7 | **a**) Submitochondrial analysis of the PDPL identified mitochondrial proteins as well as the whole dataset collected in the MitoCart3.0 database. Matrix: mitochondrial matrix; MIM: mitochondrial inner membrane; MOM: mitochondrial outer membrane; IMS: intermembrane space. The protein number identified in each sub-organelle were listed on the top.

b) Subnuclear analysis of the PDPL identified real nucleus proteins by **miniSOG-H2B** construct. Venn diagram showed the overlapping proteins in nucleoplasm, nucleus and nucleolus. **c)** Fluorescent imaging of miniSOG-3xNLS construct for its nucleus localization. Scale bar: 10 μm . **d)** Gel imaging of nucleus specific PDPL labeling by miniSOG-3xNLS. **e)** Volcano plots of PDPL-labeled proteome by miniSOG-3xNLS using label free quantification (n = 3 independent biological experiments). Significantly changed proteins are highlighted in red ($p < 0.05$ and >2 -fold ion intensity difference). Relevant proteins that are significant in HEK293T-miniSOG but not in HEK293T are marked in green. **f)** Subnuclear analysis of the PDPL identified real nucleus proteins by miniSOG-3xNLS construct. Venn diagram showed the overlapping proteins in nucleoplasm, nucleus and nucleolus. **g)** Specificity analysis for miniSOG-3xNLS derived from experiment in e. **h)** Venn diagram showing overlapping proteins in miniSOG-H2B and miniSOG-3xNLS constructs.

For the BRD4-PDPL experiments, why not also compare to the H2B-PDPL results which might be a better control for specificity?

We thank the Reviewer for this suggestion. As shown in Figure S9b, we compared the H2B-PDPL and BRD4-PDPL and found that H2B could cover around 80% of protein hits in the BRD4 result, indicating that the nucleus proteome identified by H2B could cover the more defined nucleus target, BRD4.

Line 282-283: The nuclear proteome determined by miniSOG-H2B covered 77.6% of BRD4 interacting proteins (Supplementary Fig. 9b).

Supplementary Figure 9 | **a)** Venn diagram showing overlapping proteins in C terminal and N terminal miniSOG fused BRD4. **b)** Venn diagram showing overlapping proteins in miniSOG-BRD4 and miniSOG-H2B. **c)** Volcano plots of PDPL-labeled proteomes with irradiation time: 2

min, 5 min, 10 min and 20 min (n = 3 independent biological experiments). HEK293T was used as the negative control. Significantly changed proteins are highlighted in red ($p < 0.05$ and >2 -fold ion intensity difference). Relevant proteins that are significant in HEK293T-miniSOG but not in HEK293T are marked in green. **d)** The ion intensity in label free quantification for known BRD4 binding proteins across the indicated irradiation time. **e)** Full map for string analysis of BRD4 interacting proteins with over three interactors.

Would PDPL work in the secretory pathway, in lysosomes, or on the cell surface? Perhaps just discuss.

We thank the Reviewer for this question. Yes, we believe PDPL would work in the secretory pathway, lysosome and cell surface. As APEX method has been successfully applied in the above organelles (Huang, M. et al, *ACS Chem. Biol.* **2022**, doi.org/10.1021/acscchembio.1c00865; Hao, L. et al. *Ana. Chem.* **2020**, 92, 15437-15444), PDPL would be compatible with these applications. We have discussed these possibilities in the discussion section.

Line 394-396: Proximity labeling has also been successfully applied in characterizing surfacome, lysosome proteome, and secretory pathway associated proteomes^{44, 45}. We believe PDPL would be compatible with these subcellular organelles.

Reviewer #2 (Remarks to the Author):

In the manuscript "Spatiotemporal-resolved protein networks profiling with photoactivation dependent proximity labeling", Yansheng Zhai and his colleagues engineered a photoactivated platform for characterization of protein interaction based on spatial proximity. This innovative technology has great potential in biological science.

We thank this Reviewer for their input and appreciation on our work.

The work is performed thoroughly, experiments are proved carefully.

We thank this Reviewer for this comment on our work.

I would like to discuss the following concern.

The authors state that the protein labeling reaction is based on singlet oxygen generated by miniSOG. But miniSOG is really poor singlet oxygen sensitizer ($\Delta\Phi = 0.030 \pm 0.002$) [Pimenta, F. M.; Jensen, R. L.; Breitenbach, T.; Etzerodt, M.; Ogilby, P. R. Oxygen-Dependent Photochemistry and Photophysics of "MiniSOG," a Protein-Encased Flavin. *Photochem. Photobiol.* 2013, 89, 1116–1126; Ruiz-González, R.; Cortajarena, A. L.; Mejias, S. H.; Agut, M.; Nonell, S.; Flors, C. Singlet Oxygen Generation by the Genetically Encoded Tag miniSOG. *J. Am. Chem. Soc.* 2013, 135, 9564–9567]. It is well documented that FMN encapsulated in protein scaffold produces a very limited singlet oxygen amount due to the quenching of a significant fraction of the nascent 1O_2 molecules by aromatic amino acid residues of the protein. The new form of miniSOG, SOPP3 [M.

Westberg et al. No Photon Wasted: An Efficient and Selective Singlet Oxygen Photosensitizing Protein *J. Phys. Chem. B* 2017, 121, 9366–9371] has been created which generate singlet oxygen with yield 0.6. But the authors didn't use this form during the experiments.

We thank the Reviewer for this question. We started from miniSOG as our first generation PDPL platform as miniSOG is well-established and has been used for proximal RNA labeling (Zou, P. et al, *Nat. Chem. Biol.*, **2019**, *15*, 1110-1119) and protein labeling via disulfide formation (Shu, X. et al. *Bioorg. Med. Chem. Lett.* **2016**, *26*, 3359). Moreover, the split-miniSOG has also been engineered for light and electron microscopy imaging (Ngo, J. et al. *Cell Chem. Biol.* **2019**, *26*, 1407). We will definitely optimize the system using other miniSOG-derived photo-sensitizer proteins, such as SOPP1, SOPP2 and SOPP3 in our next manuscript.

To answer this question, we have constructed the mito-SOPP3 and compared it to the mito-miniSOG. As shown in figure S11, the intensity of labeling bands increased 3-fold in the SOPP3 group, but the background labeling (lane 9, no UV) increased 3-fold at the same time. The overall signal to noise ratio didn't change between two constructs. However, we believe SOPP3 with higher singlet oxygen yield requires a shorter illumination time, which would allow for the capture of more dynamic cellular processes. We mentioned that we will focus on more efficient version of photosensitizer proteins in the future in the discussion section.

Line 413-418: A new form of miniSOG, termed SOPP3, has been engineered with improved singlet oxygen yield⁴⁷. We compared miniSOG to SOPP3 and found the labeling efficiency was increased, although the signal-to-noise remained unchanged (Supplementary Fig. 11). We posit that SOPP3 optimization (e.g., via directed evolution) would result in a more efficient photo sensitizer protein requiring a much shorter illumination time, allowing for the capture of more dynamic cellular processes.

Supplementary Figure 11 | a) Comparison of mito-miniSOG to mito-SOPP3. Both labelings could be quenched by Trolox and NaN₃. Omission of illumination or aniline probe was tested for specificity. CBB: coomassie brilliant blue. **b)** signal-to-noise ratio was evaluated based on illumination and no illumination for both proteins (miniSOG: lane 5 / lane 3; SOPP3: lane 11 / lane 9).

The only evidence that the protein labeling reaction is mediated by singlet oxygen is given in fig.1e, showing the fluorescence in mitochondria in the presence of Si-DMA probe. First, I think, this experiment lacks a negative control: HEK293 cells without miniSOG expression treated with Si-DMA. And second, what about cells treated with superoxide anion sensor (for example, MitoSOX™ Red Mitochondrial Superoxide Indicator)?

We thank the Reviewer for these two suggestions. As shown in figure S3b, we added HEK293T without miniSOG as the control for si-DMA probe. No fluorescent signal was detected across different illumination time points as expected. We also treated the cells with mitoSOX as the superoxide indicator, and no signals were detected. To make sure that mitoSOX probe is working well, an experiment where the cells were treated with doxorubicin to induce the superoxide generation was applied as the positive control.

Line 134-138: Fluorescent imaging of singlet oxygen by Si-DMA probe confirmed the presence of singlet oxygen in the HEK293T-miniSOG line, but not the parental HEK293T line. In addition, mitoSOX Red was unable to detect superoxide generation after illumination (Fig. 1e and Supplementary Fig. 3b)³⁰. These data strongly indicate singlet oxygen as the major ROS that gives rise to subsequent proteome labeling.

Supplementary Figure 3 | a) Investigation of the ROS type generated in PDPL. Trolox and sodium azide could quench the labeling, while D₂O could enhance the labeling. PDPL was insensitive to Mannitol and Vitamin C and H₂O₂ cannot trigger the labeling. **b)** Fluorescent imaging of singlet oxygen by Si-DMA probe confirmed the presence of singlet oxygen in 293T-miniSOG line but not parental 293T line. mitoSOX Red was unable to detect the superoxide after illumination. Doxorubicin was used as a positive control. Scale bar: 10 μm.

Further, the authors state that "protein labeling was greatly reduced in the presence of sodium azide, which is known to quench singlet oxygen" (p. 6, fig. d). Of course, NaN₃ an efficient quencher of ¹O₂. However, caution should be exercised with this interpretation since NaN₃ is also a very efficient quencher of the FMN triplet state [Spikes, J. D., et al. Photodynamic crosslinking of proteins. III. Kinetics of the FMN- and rose bengal-sensitized photooxidation and intermolecular crosslinking of model tyrosine-containing N-(2-hydroxypropyl)methacrylamide copolymers. Photochem. Photobiol. 1999, 70, 130–137].

As soon as miniSOG is very poor singlet oxygen sensitizer, it is worth to prove that the mechanism of proposed PDLP is indeed through singlet oxygen.

The Reviewer is correct, and the mechanism deserves further investigation. As shown in figure S3a, Trolox was also used as the singlet oxygen quencher in addition to NaN₃. Both quenchers abolished the labeling bands. Meanwhile, singlet oxygen could be stabilized in D₂O and indeed the

fluorescent bands increased accordingly. To exclude the possibility of other reactive oxygen species, H₂O₂ was added directly to the system but didn't generate the signal; Mannitol, as the hydroxyl radical scavenger, was added and didn't influence the labeling bands; Vitamin C, as the superoxide scavenger, was added and didn't quench the labeling bands either. Together, these results prove that the PDPL is mediated by singlet oxygen.

Line 129-134: Notably, protein labeling was greatly reduced in the presence of sodium azide or trolox, which are known to quench singlet oxygen²⁸. The presence of D₂O, known to stabilize singlet oxygen, enhanced labeling signal. To investigate the contribution of other reactive oxygen species to labeling, mannitol and vitamin C, established hydroxyl radical and superoxide radical scavengers respectively^{18,29}, were added but not found to reduce labeling. Addition of H₂O₂ rather than illumination failed to generate labeling (Supplementary Fig. 3a).

The manuscript can be published after major revision.

We thank the Reviewer for their support.

Reviewer #3 (Remarks to the Author):

Li and colleagues developed a novel proximity labeling method for capturing protein-protein interactions in a temporal manner from the complexed proteome samples. They fused a small photosensitizer protein, miniSOG, to the protein of interest (POI) and shed it with blue light to trigger the generation of singlet oxygen. They validated that the irradiation will oxidize histidine residues in nearby proteins which can then be labeled by an aniline-based chemical probe. The proximity-labeled proteomes can be analyzed by chemical proteomics to identify subcellular proteome composition as well as enzyme substrates. They applied the method to identify novel protein substrates of BRD4 and Parkin, and biochemically validated some of these targets. Overall, this is a well-conducted study that adds a new tool for proximity labeling other than APEX and TurboID. I support the publication of this study with some minor comments to be addressed by the authors.

We thank this Reviewer for their input on our work and support for its publication and use by the community.

- 1) The aniline probe has also been shown to label protein carbonylations in proteomes (<https://www.ncbi.nlm.nih.gov/pubmed/29569437>). Have the authors observed any labeling adducts to such modifications during their proteomic analysis?

We thank this Reviewer for this question. We re-analyzed our result and didn't observe carbonylation labeling adducts by our aniline-based probe.

- 2) Can they comment on how many histidines on average per protein are identified by the PDPL approach? Are these labeled histidines well exposed, e.g, in terms of SASA?

We thank this Reviewer for this question. We re-analyzed our data in figure 2 about the labeling sites of mitochondria proteins. As shown in figure S4, one and two histidines per protein account for over 90% of the identified sites. On average, 1.4 histidines per protein were identified by the PDPL approach. We did the SASA analysis and found the labeled histidines were indeed well-exposed.

Line 175-177: On average, 1.4 histidines per protein were identified and these labeled sites are well exposed determined by solvent-accessible surface area (SASA) and relative solvent accessibility (RSA) analysis (Supplementary Fig. 4c-d).

Supplementary Figure 4 | **a)** Evaluation of cell toxicity of PDPL. CCK-8 assay was deployed to measure the cell viability under shown conditions. **b)** pLOGO analysis of the local sequence context of PDPL-modified histidine revealed a modest motif preference for small, hydrophobic residues at the ± 1 position. **c)** Histogram plot of PDPL identified histidines per protein. **d)** Relative solvent accessibility (RSA) analysis of the identified histidine. The routinely adopted threshold 20% were labeled.

- 3) Most of the main figures are formatted with very small and over-crowding fonts, and should be reformatted for better clarity.

We thank this Reviewer for this comment. We have already enlarged the fonts and reformatted the figures for better clarity.

- 4) Figure 1a needs to be re-drawn. The amine warhead is drawn in a quite weird way. It did not convey the message that the singlet oxygen oxidizes proteins and the oxidized histone is captured then by the probe.

We agree with this Reviewer. And we have already re-drawn figure 1a.

- 5) Texts need more careful proofreading as there are several grammatical errors and poorly worded sentences throughout the manuscript.

We thank this Reviewer for this comment to help us improve the quality of the paper. We have done the proofreading for another several times and asked a native speaker to revise the manuscript.

REVIEWERS' COMMENTS

Reviewer #1 (Remarks to the Author):

The authors have sufficiently addressed my concerns with this revised manuscript and I have no additional concerns. My prior comments about the value of this manuscript remain unchanged.

Reviewer #2 (Remarks to the Author):

In this manuscript, Y. Zhai and coworkers proposed a new platform (photoactivation-dependent proximity labeling method) for characterization of protein interaction based on spatial proximity. This technology is based on genetically encoded PS miniSOG, which under blue light can generate ROS and label proteins at histidine residues. By characterizing subcellular proteomes, the authors provided evidence that the method presented is indeed works.

The manuscript has been revised thoroughly, and the authors have addressed all of my concerns. The work can be published in Nature Communications.

Reviewer #3 (Remarks to the Author):

The authors have done a great job to address the concerns and suggestions and I support its publication